# Distribution of *Helicobacter pylori* and Periodontopathic Bacterial Species in the Oral Cavity

**DOI:** 10.3390/biomedicines8060161

**Published:** 2020-06-15

**Authors:** Tamami Kadota, Masakazu Hamada, Ryota Nomura, Yuko Ogaya, Rena Okawa, Narikazu Uzawa, Kazuhiko Nakano

**Affiliations:** 1Department of Pediatric Dentistry, Osaka University Graduate School of Dentistry, Osaka 565-0871, Japan; kadota@dent.osaka-u.ac.jp (T.K.); ogaya@dent.osaka-u.ac.jp (Y.O.); rokawa@dent.osaka-u.ac.jp (R.O.); nakano@dent.osaka-u.ac.jp (K.N.); 2Department of Oral and Maxillofacial Surgery II, Osaka University Graduate School of Dentistry, Osaka 565-0871, Japan; hmdmskz@dent.osaka-u.ac.jp (M.H.); uzawa@dent.osaka-u.ac.jp (N.U.)

**Keywords:** *Helicobacter pylori*, *Porphyromonas gingivalis*, periodontopathic bacterial species, molecular biological analysis, oral specimens

## Abstract

The oral cavity may serve as a reservoir of *Helicobacter pylori*. However, the factors required for *H. pylori* colonization are unknown. Here, we analyzed the relationship between the presence of *H. pylori* in the oral cavity and that of major periodontopathic bacterial species. Nested PCR was performed to detect *H. pylori* and these bacterial species in specimens of saliva, dental plaque, and dental pulp of 39 subjects. *H. pylori* was detected in seven dental plaque samples (17.9%), two saliva specimens (5.1%), and one dental pulp (2.6%) specimen. The periodontal pockets around the teeth, from which dental plaque specimens were collected, were significantly deeper in *H. pylori*-positive than *H. pylori*-negative subjects (*p* < 0.05). Furthermore, *Porphyromonas gingivalis*, a major periodontopathic pathogen, was detected at a significantly higher frequency in *H. pylori*-positive than in *H. pylori*-negative dental plaque specimens (*p* < 0.05). The distribution of genes encoding fimbriae (*fimA*), involved in the periodontal pathogenicity of *P. gingivalis*, differed between *H. pylori*-positive and *H. pylori*-negative subjects. We conclude that *H. pylori* can be present in the oral cavity along with specific periodontopathic bacterial species, although its interaction with these bacteria is not clear.

## 1. Introduction

Evidence indicates that *Helicobacter pylori* infects the human organism via the oral cavity and subsequently remains in the gastric tissue for the rest of its host’s life [1]. Chronic inflammation of the gastric mucosa is induced by ammonia and toxins produced by *H. pylori* [1], eventually causing gastric mucosal damage and diseases such as peptic ulcer and gastric cancer. *H. pylori* in the oral cavity may cause reinfection of the stomach after eradication therapy [2,3]. However, *H. pylori* colonizes the oral cavity through an unknown mechanism.

Among the more than 700 bacterial species that reside in the human oral cavity [4], some contribute to the progression of periodontal disease. For example, *Porphyromonas gingivalis*, *Treponema denticola*, and *Tannerella forsythia* are called the red complex because of their high pathogenicity [5]. The orange complex comprises bacteria such as *Prevotella intermedia*, *Prevotella nigrescens*, and *Campylobacter rectus*, which are associated with periodontal disease, and provides a link between pathogenic species such as the red complex and commensal bacteria [5]. Furthermore, commensal bacteria known as early colonizers with low pathogenicity include species in the green and yellow complexes, such as *Capnocytophaga ochracea*, *Capnocytophaga sputigena*, and *Eikenella corrodens*. These latter species are not directly involved in the development of periodontal disease, although they are involved in the establishment of the red and orange complexes [5].

A large number of studies have shown that periodontal bacteria are associated with various systemic diseases, such as diabetes, cardiovascular disease, aspiration pneumonia, preterm birth, low birth weight, and Alzheimer′s disease [6]. Recent studies have shown differences in the oral microbiota between cancer patients and healthy subjects [7], and that specific oral microbiota may be involved in the development of cancers other than in the head and neck region. Alterations in the oral microbiota tend to be remarkable in cancer patients, especially in gastrointestinal tumor patients [7], and some epidemiological studies have pointed to a relationship between periodontal disease and the risk of gastric cancer [7]. Nevertheless, few studies have focused on the relationship between the presence *H. pylori* in the oral cavity and that of periodontopathic bacterial species [8].

The presence of small numbers of the keystone pathogen *P. gingivalis* significantly influences the development of periodontal disease [9]. An important pathogenic factor of *P. gingivalis* is an approximately 41 kDa filamentous appendage (FimA), encoded by the genes *fimA*, which is expressed on the cell surface. The *fimA* genes are classified as genotypes I to V and Ib [10]. The *fimA* genotypes I, III, and V are mainly detected in healthy gingival tissue, whereas types II, IV, and Ib are mainly detected in periodontal tissues of patients with periodontitis [10].

The detection of *H. pylori* in the oral cavity, which is widely performed using PCR, ranges from 0% to 100% [11]. These results indicate that it is difficult to detect *H. pylori* in the oral cavity with high sensitivity and specificity, because many bacterial species are present. To overcome this problem, we synthesized novel PCR primer sets using the consensus sequences of the genes of 50 strains of *H. pylori* [12]. Nested PCR analyses using these primers detect at least 1 to 10 *H. pylori* cells in the oral cavity [13]. Furthermore, closely related *H. pylori* species are not detected using this method [13]. We found that this novel nested PCR method detects *H. pylori* in samples of oral cavity tissues acquired from subjects of all ages [12,13,14,15,16].

Periodontal disease is associated with the colonization of the oral cavity by *H. pylori* [17,18]. However, most studies have focused on clinical periodontal conditions, whereas there are few studies considering the distribution of periodontopathic bacterial species. Here, we examined oral cavity specimens to analyze the association between *H. pylori* and periodontal disease, focusing on major periodontopathic bacterial species.

## 2. Materials and Methods

### 2.1. Ethics Statement

This study was conducted in compliance with the Declaration of Helsinki. The Ethics Committee of Osaka University Graduate School of Dentistry approved this study (Approval number: H30-E32, 4 December 2018). Before specimen collection, all subjects provided written informed consent.

### 2.2. Subjects

The subjects (13 males and 26 females; age range, 16–70 years; median age, 31 years; mean age, 35.3 ± 15.1 years) were referred to the Department of Oral and Maxillofacial Surgery at Osaka University Dental Hospital from January 2019 to February 2020 because of problems requiring tooth extraction of the third molar, such as dental caries and pericoronitis. Sixteen of 39 extracted teeth were impacted. Patients’ characteristics were as follows: history of systemic and gastrointestinal diseases, prior *H. pylori* infection of gastric tissue, and prior eradication of *H. pylori* from gastric tissue. The presence or absence of dental caries in the extracted tooth was diagnosed by visual inspection, palpation with a dental explorer, and X-ray examination. The depth of the periodontal pocket (also called probing depth) was measured from the gingival margin to the bottom of the pocket using a dental probe. The deepest level reached was recorded. Extraction of the third molar, medical examination by an interview, and measurement of the depth of the periodontal pockets were performed by a single doctor in the department of oral and maxillofacial surgery, and dental caries diagnosis was performed by a single dentist in the department of pediatric dentistry.

### 2.3. Oral Cavity Specimens

The extracted third molar was placed in a sterile plastic tube containing 2.5 mL of sterile saline, and sonication was used to separate the dental plaque from the tooth. After the tooth was removed, the suspension was centrifuged, and the supernatant was discarded. The resulting suspension served as a dental plaque specimen from the extracted tooth. Next, the pulp chamber of the extracted tooth was opened using a sterilized dental handpiece and diamond point, and the dental pulp specimen was deposited into a sterile plastic tube containing 1 mL of sterile saline, according to a published method [12]. Saliva (1 mL) from each subject was collected into a sterile disposable tube. The dental pulp and saliva specimens were centrifuged, and the supernatant was discarded. Bacterial DNA was extracted from these specimens of dental plaque, dental pulp, and saliva as described below.

### 2.4. H. pylori Strains and Growth Condition

*H. pylori* reference strain J99 (ATCC 700824) purchased from Summit Pharmaceuticals International corporation (Tokyo, Japan) served as the positive control. Blood agar plates (Becton Dickinson, Franklin Lakes, NJ, USA), incubated at 37 °C for 3–5 days, were used to isolate bacterial colonies [12]. Thereafter, colonies were inoculated into 10 mL of brucella broth (Becton Dickinson) supplemented with 1 mL of horse serum by using a sterilized platinum loop and then incubated at 37 °C for 24 h under microaerophilic conditions. The bacteria were collected using centrifugation at 8000 rpm for 10 min, and genomic DNA was extracted as described below.

### 2.5. Bacterial DNA Extraction

Bacterial DNA was extracted using a published method [12]. Briefly, oral specimens or *H. pylori* strain J99 were resuspended in 250 μL of 10 mM Tris-HCl (pH 8.0) containing 100 mM NaCl and 1 mM EDTA. The cells were collected using centrifugation, lysed in 600 µL of Cell Lysis Solution (Qiagen, Düsseldorf, Germany), and incubated at 80 °C for 5 min, followed by the addition of 3 μL of RNase A (10 mg/mL; Qiagen) and incubation at 37 °C for 30 min. Protein Precipitation Solution (Qiagen) was added, and the mixture was vigorously vortexed for 20 s and then centrifuged at 10,000× *g* for 3 min. The supernatant was combined with 600 μL of isopropanol (Wako Pure Chemical Industries, Tokyo, Japan) and centrifuged. The precipitate was then resuspended in 70% ethanol (Wako Pure Chemical Industries), centrifuged, combined with 100 μL of 100 µL TE buffer (10 mM Tris-HCl, 1 mM EDTA (pH 8.0)), and used in the next study.

### 2.6. PCR Detection of H. pylori

Nested PCR was performed using bacterial DNA extracted from oral cavity specimens with *H. pylori*-specific primer sets described in our previous study [13] (Table 1). Briefly, PCR assays to detect *H. pylori* were first performed using the primers *ureA*-aF and *ureA*-bR, followed by a second step using the primers *ureA*-bF and *ureA*-aR. For first-step PCR, 2 μL of bacterial DNA extracted from oral cavity specimens was amplified in 20 μL of reaction mixture. For second-step PCR, 1 μL of the first PCR product was used as a template for 20 μL reactions containing TaKaRa Ex *Taq* polymerase (Takara Bio. Inc., Otsu, Japan). In the first- and second-step reactions, PCR amplification was performed as follows: initial denaturation at 95 °C for 4 min, followed by 30 cycles at 95 °C for 30 s, 55 °C for 30 s, 72 °C for 30 s, and final extension at 72 °C for 7 min. The PCR products were fractionated using a 1.5% (*w*/*v*) agarose gel containing Tris-acetate-EDTA buffer, stained with ethidium bromide (0.5 μg/mL), and visualized under UV illumination.

### 2.7. PCR Detection of Periodontopathic Bacterial Species

PCR was performed to detect the following periodontitis-related bacterial species: *P. gingivalis*, *T. denticola*, *T. forsythia*, *C. ochracea*, *C. sputigena*, *P. intermedia*, *P. nigrescens*, *C. rectus*, *Aggregatibacter actinomycetemcomitans*, and *E. corrodens*. A ubiquitous primer set which matches almost all bacterial 16S rRNA genes was used as a positive control [19]. For PCR, 2 μL of bacterial DNAs was amplified using the respective species-specific primer sets and TaKaRa Ex Taq polymerase [20,21,22,23,24] (Table 1). PCR assays were performed using the following cycling parameters: initial denaturation at 95 °C for 4 min, followed by 30 cycles at 95 °C for 30 s, 58 °C for 30 s, and 72 °C for 30 s, with a final extension at 72 °C for 7 min. The PCR products were fractionated using a 1.5% (*w*/*v*) agarose gel containing Tris-acetate-EDTA buffer, stained with ethidium bromide (0.5 μg/mL), and visualized under UV illumination.

### 2.8. PCR Detection of fimA Genotypes

The *fimA* genotypes of *P. gingivalis* were determined using *fimA* primers specific for types I to V and type Ib genotypes [25,26,27]. (Table 1). For PCR, 2 μL of bacterial DNAs was amplified using the respective species-specific primer sets and TaKaRa Ex *Taq* polymerase. PCR assays were performed using the following cycling parameters: initial denaturation at 95 °C for 4 min, followed by 30 cycles at 95 °C for 30 s, 58 °C for 30 s, and 72 °C for 30 s, with a final extension at 72 °C for 7 min. The PCR products were fractionated using a 1.5% (*w*/*v*) agarose gel containing Tris-acetate-EDTA buffer, stained with ethidium bromide (0.5 μg/mL), and visualized under UV illumination.

### 2.9. Statistical Analysis

Statistical analyses were performed using GraphPad Prism 6 (GraphPad Software Inc., La Jolla, CA, USA). Comparisons between two groups were performed using the Student’s *t* test. Intergroup differences of each analysis were determined using analysis of variance. The Bonferroni correction was used for post-hoc analysis; *p* < 0.05 was considered statistically significant.

## 3. Results

### 3.1. Clinical Characteristics of H. pylori-Positive and H. pylori-Negative Subjects

Saliva was collected from 39 subjects who underwent extraction of the third molar. Dental plaque and pulp specimens were collected from the extracted teeth. Nested PCR was performed using as a template bacterial DNA obtained from saliva, extracted teeth, and dental pulp specimens, using the previously described *H. pylori*-specific primers sets [13]. *H. pylori* was detected in 10 of 39 samples (25.9%), which were defined as *H. pylori*-positive (Table 2, Figure 1A). Interestingly, 16 out of 39 specimens were impacted teeth, three (18.8%) of which were positive for *H. pylori*. The percentage of subjects with history of systemic and gastrointestinal diseases, prior *H. pylori* infection, and previous eradication of *H. pylori* was higher for *H. pylori*-positive subjects than for *H. pylori*-negative subjects, although the differences were not significant (Table 2).

### 3.2. Detection of H. pylori and Periodontopathic Bacterial Species

*H. pylori* was detected in 7 of 39 dental plaque specimens (17.9%), while only 2 of 39 saliva (5.1%) and 1 of 39 dental pulp specimens (2.6%) were positive for *H. pylori* (Figure 1B). Furthermore, 10 major periodontopathic bacterial species were detected in oral cavity specimens using a PCR method with species-specific sets of primers (Table 1). Among 10 major periodontal pathogens, the prevalence of *P. gingivalis* in all 39 specimens was 33.3% (*n* = 13) and ranged between 10.3% (*n* = 4) and 15.4% (*n* = 6) for each type of specimen (Figure 1A,B). The rates of detection of *T. forsythia*, *P. nigrescens*, *C. rectus*, *C. ochracea*, *C. sputigena*, and *E. corrodens* ranged between 74.4% (*n* = 29) and 89.7% (*n* = 35) among all 39 oral cavity specimens. These bacterial species were detected at the highest rates in dental plaque. In all specimens, *T. denticola* and *P. intermedia* were detected at lower rates, and *A. actinomycetemcomitans* was not detected in any specimen. When the distribution of bacterial species in each oral specimen was analyzed, *H. pylori* was detected in 7 of 10 (70%) dental plaque samples (Figure 1C). *P. gingivalis* was detected in 6 of 13 dental pulp (46.2%) samples, as well as in saliva (*n* = 2), dental plaque (*n* = 3), and both saliva and dental plaque specimens (*n* = 2). *T. denticola* (5 of 6; 83.3%) and *C. ochracea* (20 of 30; 66.7%) were abundant in dental plaque. *T. forsythia* (17 of 35; 48.6%), *P. nigrescens* (19 of 34; 55.9%), *C. rectus* (18 of 35; 51.4%), *C. sputigena* (13 of 31; 41.9%), and *E. corrodens* (19 of 36; 52.8%) were detected in multiple oral specimens in approximately 50% of subjects.

### 3.3. Relationship between H. pylori and Oral Disease

*H. pylori* was detected mainly in the dental plaque around the extracted teeth. Therefore, we analyzed the relationship between the presence of *H. pylori* and major oral diseases affecting the extracted teeth, such as dental caries, as well as periodontal pocket depth. In subjects with dental caries, those with *H. pylori* were more numerous than those without *H. pylori*, although the difference was not significant (Appendix A). Furthermore, periodontal pockets of 23 of the 39 erupted teeth were measured, because the remaining 16 extracted teeth were impacted. Thus, the average periodontal pocket depths of *H. pylori*-positive and -negative specimens were 3.8 mm and 3.1 mm, respectively (*p* < 0.05) (Figure 2A). Similarly, there was a significant difference in average periodontal pocket depth between *P. gingivalis*-positive and -negative specimens (*p* < 0.01) (Figure 2B). In contrast, *C. rectus* and *C. sputigena* were more frequent in the periodontal pocket among specimens without detectable bacteria (*p* < 0.05) (Figure 2C,D). Furthermore, no significant differences between the presence or the absence of other periodontopathic bacterial species were found (Figure 2E–G). There were few *P. intermedia*-positive and *A. actinomycetemcomitans*-positive specimens and few *P. nigrescens*-negative and *E. corrodens*-negative specimens in the extracted teeth. Therefore, these specimens were not suitable for the analysis (data not shown). Furthermore, 23 subjects with a periodontal pocket of 3 mm or less were classified as healthy, and those with a pocket of 4 mm or more were classified as having periodontitis, as defined in a previous study [28]. As a result, only *H. pylori* and *P. gingivalis* showed high rates of detection in the periodontitis group, although there were no significant differences between the groups (Appendix A).

### 3.4. Relationship between H. pylori and Periodontopathic Bacterial Species

We evaluated the relationship between the presence of *H. pylori* and the distribution of periodontopathic bacterial species in the oral cavity. At least four periodontopathic bacterial species were detected in most dental plaque specimens, with or without *H. pylori*, although a few were found in certain *H. pylori*-negative subjects (Figure 3A).

We next classified the periodontopathic bacterial species into groups according to decreasing pathogenicity [5] as follows: red complex (*P. gingivalis*, *T. denticola*, and *T. forsythia*), orange complex (*P. intermedia*, *P. nigrescens*, and *C. rectus*), and green complex (*C. ochracea*, *C. sputigena*, *A. actinomycetemcomitans*, and *E. corrodens*). The number of red-complex species in *H. pylori*-positive subjects was significantly higher than that in *H. pylori*-negative subjects (*p* < 0.01) (Figure 3B). In contrast, the number of orange complex species in *H. pylori*-positive subjects was significantly lower than that in *H. pylori*-negative subjects (*p* < 0.05) (Figure 3C); the results were similar for the green complex (Figure 3D).

We next selected two strains of the red complex and assessed their relationship with *H. pylori*. We found that *P. gingivalis* and *T. denticola* were detected at significantly higher frequencies in *H. pylori*-positive specimens compared with *H. pylori*-negative specimens. (*p* < 0.05) (Figure 3E). Furthermore, there was a significant association with the combination of *P. gingivalis* and *T. forsythia* (Figure 3F) and with the combination of *T. denticola* and *T. forsythia* (Figure 3G).

The relationships between the presence or absence of *H. pylori* and the detection frequency of each periodontopathic bacterial species are shown in Figure 3H. The detection rates of *P. gingivalis*, *T. forsythia*, and *P. intermedia* in *H. pylori*-positive subjects was higher than that in *H. pylori*-negative subjects. Among these bacterial species, 3 of 7 *H. pylori*-positive specimens were *P. gingivalis*-positive (42.9%), which was significantly higher than for *H. pylori*-negative specimens (2 of 32; 6.3%) (*p* < 0.05). In contrast, *C. ochracea*, *C. sputigena*, and *C. rectus* were detected at lower rates in *H. pylori*-positive specimens compared with *H. pylori*-negative specimens, although the differences were not significant.

### 3.5. Distribution of fimA Genotypes in Subjects with or without H. pylori

The detection rate of *P. gingivalis* was higher in *H. pylori*-positive subjects. We therefore analyzed *P. gingivalis*-positive subjects, focusing on the distribution of *fimA* genotypes. Six of eight *H. pylori*-negative subjects were classified as type IV *fimA* (Figure 4A). In contrast, type IV *fimA* was detected in one of five *H. pylori*-positive subjects, and types I and II *fimA* genotypes were detected in four and three *H. pylori*-positive subjects, respectively (Figure 4B). When the distribution frequencies of *fimA* genotypes with or without *H. pylori* were compared, type II *fimA* was detected at a significantly higher frequency in *H. pylori*-positive subjects (Figure 4C). Although type I *fimA* and type Ib *fimA* genotypes were detected at high frequencies in *H. pylori*-positive subjects, no significant difference with respect to *H. pylori*-negative subjects was found. In contrast, the type IV *fimA* genotype was detected at high frequency in *H. pylori*-negative subjects, although no significant difference with respect to *H. pylori*-positive subjects was found.

## 4. Discussion

*H. pylori*, a Gram-negative microaerophilic bacterium, causes gastric diseases and may be transmitted via the oral cavity [1]. A relationship is considered to exist between the presence of *H. pylori* in the oral cavity and the severity of periodontal disease [17,18]. Previous studies mainly focused on the relationship between *H. pylori* infection and clinical periodontal status by comparing the distribution frequency of *H. pylori* in healthy subjects with that in periodontal disease patients [17,18]. However, differences in the periodontopathic bacterial species between *H. pylori*-positive and *H. pylori*-negative subjects in the oral cavity remain unknown. Therefore, we investigated the relationship between the presence of *H. pylori* in the oral cavity and periodontal disease by focusing on the distribution of periodontopathic bacterial species.

Recent reviews suggest that periodontopathic bacterial species may affect human health [6,29]. Highly pathogenic bacteria involved in periodontal disease, such as *P. gingivalis*, can also be associated with systemic diseases such as diabetes and arteriosclerosis [6]. Interestingly, abnormal oral microbiota or periodontal disease are also associated with an increased risk of gastric cancer [7,30]. Nevertheless, the periodontopathic bacterial species that co-exist with *H. pylori* in the oral cavity remain unknown. In the present study, we were able to detect not only species that confer periodontal disease but also *H. pylori* in the oral cavity and to clarify the relationship between these bacteria. If the coexistence of specific periodontopathic bacterial species and *H. pylori* can be shown to be associated with gastric disease, these bacterial species may be used as diagnostic markers for gastric disease or as targets for therapeutic interventions.

Although most methods used to detect *H. pylori* in the oral cavity employ PCR assays, it is difficult to detect *H. pylori* with high specificity and high sensitivity in the presence of approximately other 700 oral bacterial species [4,13]. Therefore, we developed a reliable nested PCR method based on the complete genome sequences of approximately 50 *H. pylori* strains [12,13]. Using this method, we accurately detected *H. pylori* in different types of oral specimens. Similarly, we used a PCR method to detect periodontopathic bacterial species in clinical specimens [31,32,33]. No previous study has attempted to detect both *H. pylori* and periodontopathic bacterial species at the same time using these molecular analyses. Here, we used methods from previous studies to determine the effects of periodontopathic bacterial species on the colonization of the oral cavity by *H. pylori*.

We used a novel nested PCR method to detect *H. pylori* in saliva, dental plaque, and dental pulp. We found that *H. pylori* was detected at the highest frequency in dental plaque. This result is consistent with previous reports showing that dental plaque harbors *H. pylori* [34,35]. In contrast, *H. pylori* was less frequent in saliva, which is consistent with our previous study [12]. Although only a few studies have focused on *H. pylori* in dental pulp specimens [12,16,36], dental pulp may serve as a reservoir for *H. pylori* only when the pulp is exposed by dental caries or traumas. Here, more than half of the extracted teeth did not exhibit dental caries, and only a few had severe dental caries extending to the pulp, likely explaining the low detection frequency of *H. pylori*. These results suggest that *H. pylori* does not easily enter the dental pulp, unless it is exposed because of a dental problem.

Periodontal pockets are caused by the inflammatory response to periodontopathic bacteria in the dental plaque accumulated in periodontal tissues [37]. A deep periodontal pocket has a low oxygen concentration and is hemorrhagic [37,38], which accelerates the growth of periodontopathic bacterial species. In the present study, periodontal pockets were significantly deeper in *H. pylori*-positive specimens. Among the 11 bacterial species studied, only *H. pylori* and *P. gingivalis* were associated with other bacteria and increased periodontal pocket depth. Therefore, the environment of the periodontal pocket may favor the colonization by *H. pylori*. Furthermore, *H. pylori* may be involved in the formation of periodontal pockets. *H. pylori* as well as *Campylobacter* species are major microaerobic periodontopathic bacteria [39]. Therefore, periodontal pockets with low oxygen concentrations may serve as reservoirs of *H. pylori*.

Pericoronitis is a gingival inflammation caused by bacterial infection of the periodontal tissues surrounding tilted or impacted third molar or unerupted teeth [40]. Thus, we detected *H. pylori* from third molar that was diagnosed as impacted. As a result, *H. pylori* was detected in 3 of 16 (18.8%) impacted teeth. Therefore, impacted third molar may be a possible reservoir for *H. pylori* colonization.

To the best of our knowledge, only one study has investigated the correlation between the presence of *H. pylori* and that of periodontopathic bacterial species in the oral cavity, examining 14 Chinese subjects [8]. In this study, the detection rates of three of five periodontopathic bacterial species, including *P. gingivalis*, were increased in *H. pylori*-positive dental plaque specimens. By referring to this paper, we decided to analyze three samples consisting of dental plaque, saliva, and dental pulp from 39 subjects. However, the sample sizes in both the Chinese study and our present study are small. Larger-scale studies with greater numbers of such subjects as well as other populations should be performed in the future, because half of the world’s population is infected by *H. pylori* [41].

In addition to the known associations of oral bacterial species with diabetes and arteriosclerosis, recent analyses of oral bacteria, including periodontopathic bacteria, have led to important findings related to systemic diseases, such as gastric cancer [7]. Many of these findings have come from studies with small sample sizes; however, systematic reviews and meta-analyses based on these papers have indicated possible relationships [7]. Therefore, we consider that our research has the potential to lead to new clinical findings.

Periodontal disease is caused by diverse periodontopathic bacterial species, which are classified according to differences in pathogenicity determined using molecular biological analysis [5]. The most pathogenic group is the red complex composed of *P. gingivalis*, *T. denticola*, and *T. forsythia*, which mainly colonizes the adult oral cavity [5]. Here, we showed that *H. pylori*-positive specimens contained multiple red-complex members that are involved in coaggregation or use mutual metabolites to enhance their pathogenicity [42,43,44]. Furthermore, immunohistochemical analyses showed that red-complex bacteria coexist in dental plaque [45].

*H. pylori* may be involved in a symbiotic relationship with these bacterial species. Although deep periodontal pockets may represent an important factor required for colonization of the oral cavity by *H. pylori*, the detection of *H. pylori* correlated with the presence of red-complex species such as *T. denticola* and *T. forsythia*, which are not involved in the formation of periodontal pockets. Therefore, the presence of these bacteria in the absence of periodontal pockets may represent a risk factor for *H. pylori* colonization.

Members of the orange complex such as *P. intermedia* and *P. nigrescens*, which are present during adolescence, act as a link between the red complex and commensal oral bacteria [5]. Members of the green and yellow complexes, including *C. ochracea* and *C. sputigena*, are commensal species with low pathogenicity [5]. These commensal bacteria are established during childhood in the absence of periodontal disease, which may be required for the establishment of the red and orange complexes. Interestingly, we frequently detected members of the orange and green complexes in specimens without *H. pylori*. Therefore, the presence of commensal bacteria with low periodontal pathogenicity suppress the colonization of the oral cavity by *H. pylori*.

The detection rate of *P. gingivalis* was significantly higher in *H. pylori*-positive subjects than in the *H. pylori*-negative subjects. However, the presence or absence of *H. pylori* did not affect the detection rate of other periodontopathic bacterial species. Therefore, we further analyzed the relationship between the presence or absence of *H. pylori* and the *fimA* genotypes of *P. gingivalis*. In general, genotypes II and IV are found in oral cavity specimens collected from patients with severe periodontal disease [10], and genotype I is found in oral samples of healthy individuals [10]. Here, we showed that genotypes I and II were frequently detected in *H. pylori*-positive specimens, and genotype IV was frequently detected in *H. pylori*-negative specimens. Thus, *P. gingivalis* with a specific *fimA* genotype may be involved in the colonization by *H. pylori*, which is not necessarily highly virulent in periodontal disease.

This study has certain limitations. First, the number of specimens was small, although we established a relationship between *H. pylori* and specific periodontopathic bacterial species. In addition, the incidences of *H. pylori* and many periodontopathic bacterial species increase with age. To overcome these limitations, larger scale studies with more subjects and samples taken from various teeth are needed. Furthermore, we were unable to determine if *H. pylori* was present in the periodontal pocket, because the dental plaque specimens were collected from around the extracted teeth. Thus, future studies should compare the detection rates of *H. pylori* in dental plaques collected from different parts of a tooth. Moreover, whether *H. pylori* can affect periodontal disease is unknown. *P. gingivalis* is a keystone pathogen that induces immunological abnormalities followed by a drastic increase in the population of oral bacteria that cause the damage of periodontal disease [9]. Thus, using *H. pylori*, *P. gingivalis* creates an oral microbiome more favorable for its own growth. In contrast, one hypothesis states that *H. pylori* may change the subgingival flora that contributes to the exacerbation of periodontal disease [8].

In summary, we frequently detected *H. pylori* in dental plaque around an extracted tooth. Furthermore, *H. pylori* was frequently detected in dental plaque specimens with deep periodontal pockets and characterized by the presence of by *P. gingivalis* with a specific *fimA* genotype. *H. pylori* was also detected in dental plaque specimens with multiple periodontopathic bacterial species of the red complex, even when the periodontal pocket was shallow. Taken together, these results suggest that *H. pylori* can co-exist with specific periodontopathic bacterial species, although interactions among the bacteria are not clear.

## Figures and Tables

**Figure 1 biomedicines-08-00161-f001:**
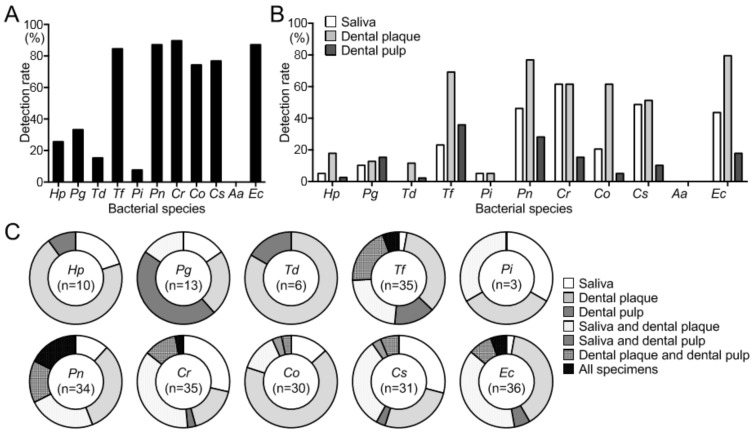
Distribution of *H. pylori* and periodontopathic bacterial species in the oral cavity of the study subjects (*n* = 39). (**A**) Rates of detection of bacterial species among all oral specimens. (**B**,**C**) Rates of detection of each bacterial species in the oral specimens. *Hp*, *H. pylori*; *Pg*, *P. gingivalis*; *Td*, *T. denticola*; *Tf*, *T. forsythia*; *Pi*, *P. intermedia*; *Pn*, *P. nigrescens*; *Cr*, *C. rectus*; *Co*, *C. ochracea*; *Cs*, *C. sputigena*; *Aa*, *A. actinomycetemcomitans*; and *Ec*, *E. corrodens*.

**Figure 2 biomedicines-08-00161-f002:**
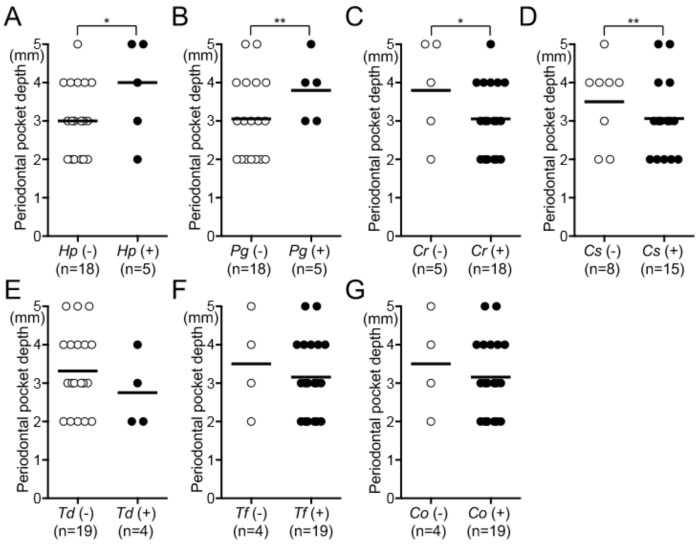
Comparisons of detection frequencies of *H. pylori* and periodontopathic bacteria in the periodontal pockets of the extracted teeth. Periodontal pocket depth around the extracted teeth with or without (**A**) *H. pylori*, (**B**) *P. gingivalis*, (**C**) *C. rectus*, (**D**) *C. sputigena*, (**E**) *T. denticola*, (**F**) *T. forsythia*, and (**G**) *C. ochracea*. Significant differences, * *p* < 0.05 and ** *p* < 0.01. Black lines indicate mean values of the respective groups. *Hp*, *H. pylori*; *Pg*, *P. gingivalis*; *Cr*, *C. rectus*; *Cs*, *C. sputigena*; *Td*, *T. denticola*; *Tf*, *T. forsythia*; and *Co*, *C. ochracea*.

**Figure 3 biomedicines-08-00161-f003:**
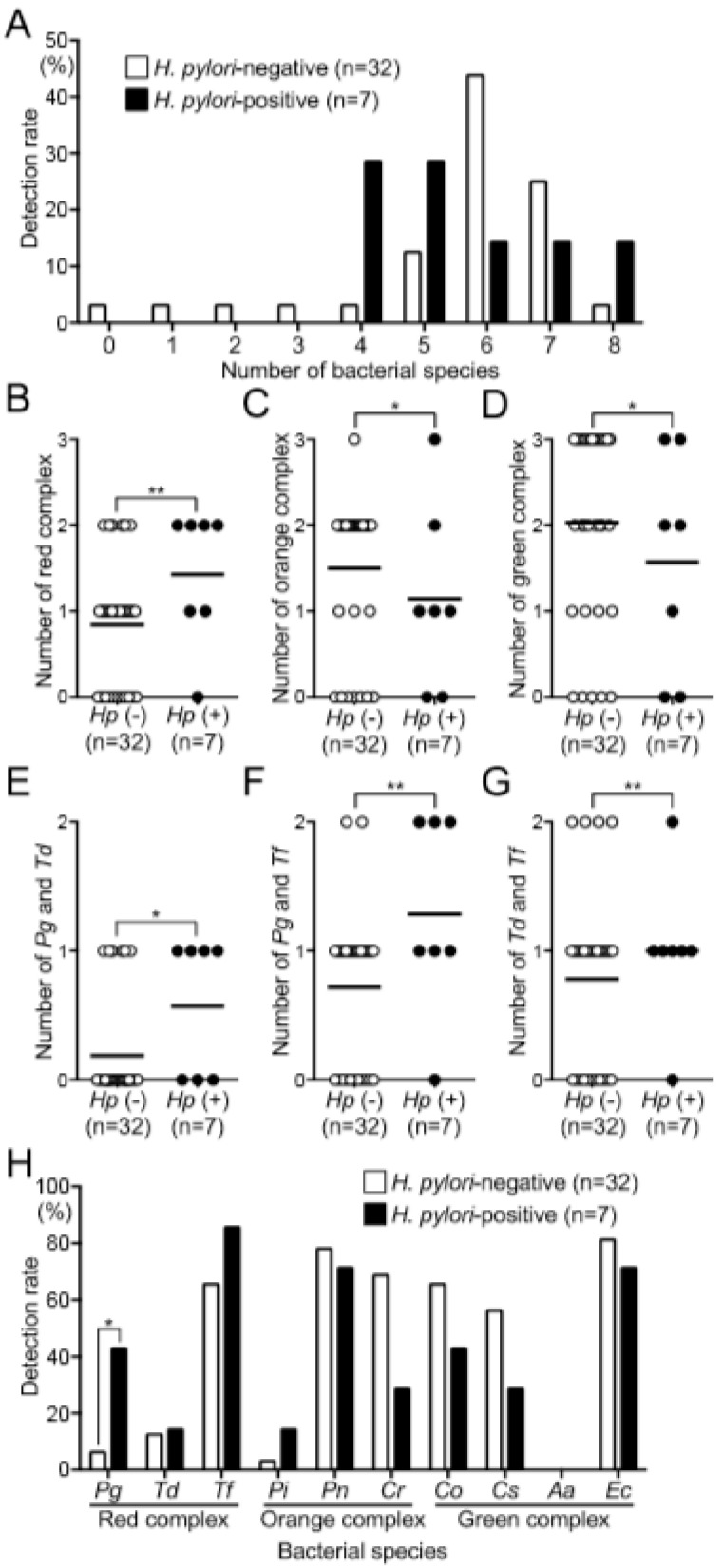
Relationship between the presence or absence of *H. pylori* and the detection of periodontopathic bacterial species in the extracted teeth. (**A**) Distributions of periodontopathic bacterial species associated with *H. pylori*. Bacterial species of the (**B**) red complex (*P. gingivalis*, T. *denticola*, and *T. forsythia*), (**C**) orange complex (*P. intermedia*, *P. nigrescens*, and *C. rectus*), and (**D**) green complex (*C. ochracea*, *C. sputigena*, *A. actinomycetemcomitans*, and *E. corrodens*). Numbers of bacterial species associated with two pairs of red complex species: (**E**) *P. gingivalis* and *T. denticola*, (**F**) *P. gingivalis* and *T. forsythia*, and (**G**) *T. denticola* and *T. forsythia*. Black lines in (**B**–**G**) indicate mean values of the respective groups. (**H**) Distributions of periodontopathic bacterial species. Significant differences, * *p* < 0.05 and ** *p* < 0.01.

**Figure 4 biomedicines-08-00161-f004:**
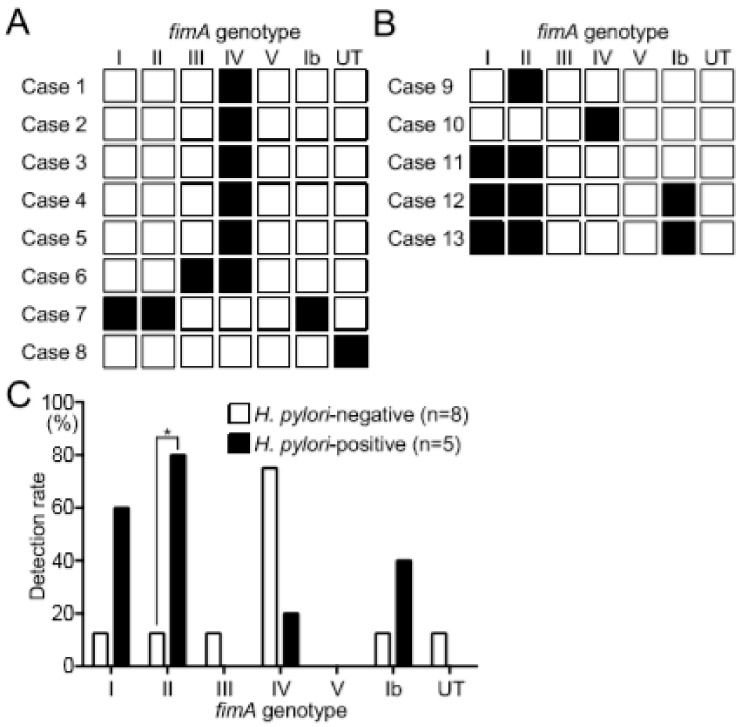
Distributions of *fimA* genotypes in subjects with and without *H. pylori*. Distributions of *fimA* genotypes in *H. pylori*-negative (**A**) and *H. pylori*-positive subjects (**B**). Black and white squares indicate positive and negative detection of *H. pylori*, respectively. (**C**) Comparisons of *fimA* genotypes of *H. pylori*-positive and *H. pylori*-negative subjects. Significant differences, * *p* < 0.05.

**Table 1 biomedicines-08-00161-t001:** Polymerase chain reaction primers.

Purpose	Sequence (5′-3′)	Size (bp)	References
Universal primer			
(positive control)		315	[19]
PA (forward)	F: AGA GTT TGA TCC TGG CTC AG		

PD (reverse)	R: GTA TTA CCG CGG CTG CTG		
Detection of *H. pylori*			
First step PCR		488	[13]
*ureA*-aF	F: ATG AAA CTC ACC CCA AAA GA		
*ureA*-bR	R: CCG AAA GTT TTT TCT CTG TCA AAG TCT A		
Second step PCR		383	[13]
*ureA*-bF	F: AAA CGC AAA GAA AAA GGC ATT AA		
*ureA*-aR	R: TTC ACT TCA AAG AAA TGG AAG TGT GA		
Detection of periodontitis-related species		267	[20]
*Porphyromonas gingivalis*	F: CCG CAT ACA CTT GTA TTA TTG CAT GAT A		
	R: AAG AAG TTT ACA ATC CTT AGG ACT GTC T		
*Treponema denticola*	F: AAG GCG GTA GAG CCG CTC A	311	[21]
	R: AGC CGC TGT CGA AAA GCC CA		
*Tannerella forsythia*	F: GCG TAT GTA ACC TGC CCG CA	641	[22]
	R: TGC TTC AGT GTC AGT TAT ACC T		
*Capnocytophaga ochracea*	F: AGA GTT TGA TCC TGG CTC AG	185	[23]
	R: GAT GCC GTC CCT ATA TAC TAT GGG G		
*Capnocytophaga sputigena*	F: AGA GTT TGA TCC TGG CTC AG	185	[23]
	R: GAT GCC GCT CCT ATA TAC CAT TAG G		
*Prevotella intermedia*	F: TTT GTT GGG GAG TAA AGC GGG	575	[22]
	R: TCA ACA TCT CTG TAT CCT GCG T		
*Prevotella nigrescens*	F: ATG AAA CAA AGG TTT TCC GGT AAG	804	[22]
	R: CCC ACG TCT CTG TGG GCT GCG A		
*Campylobacter rectus*	F: TTT CGG AGC GTA AAC TCC TTT TC	598	[22]
	R: TTT CTG CAA GCA GAC ACT CTT		
*Aggregatibacter actinomycetemcomitans*	F: CTA CTA AGC AAT CAA GTT GCC C	262	[24]
	R: CCT GAA ATT AAG CTG GTA ATC		
*Eikenella corrodens*	F: CTA ATA CCG CAT ACG TCC TAA G	688	[22]
	R: CTA CTA AGC AAT CAA GTT GCC C		
Specification of *fimA* genotype		392	[25]
Type I *fimA*	F: CTG TGT GTT TAT GGC AAA CTT C		
	R: AAC CCC GCT CCC TGT ATT CCG A		
Type II *fimA*	F: ACA ACT ATA CTT ATG ACA ATG G	257	[25]
	R: AAC CCC GCT CCC TGT ATT CCG A		
Type III *fimA*	F: ATT ACA CCT ACA CAG GTG AGG C	247	[25]
	R: AAC CCC GCT CCC TGT ATT CCG A		
Type IV *fimA*	F: CTA TTC AGG TGC TAT TAC CCA A	251	[25]
	R: AAC CCC GCT CCC TGT ATT CCG A		
Type V *fimA*	F: AAC AAC AGT CTC CTT GAC AGT G	462	[26]
	R: TAT TGG GGG TCG AAC GTT ACT GTC		
Type Ib *fimA*	F: CAG CAG AGC CAA AAA CAA TCG	271	[27]
	R: TGT CAG ATA ATT AGC GTC TGC		

F; forward primer, R; reverse primer.

**Table 2 biomedicines-08-00161-t002:** Clinical characteristics of patients with and without detectable *H. pylori* sequences.

Clinical Characteristics	*H. pylori*-Negative (*n* = 29)	*H. pylori*-Positive (*n* = 10)
Age (years; mean ± SD)	32.1 ± 12.9	44.4 ± 17.9
Sex (Male (%))	10 (34.5%)	3 (30.0%)
History of systemic disease (%)	8 (27.6%)	5 (50.0%)
History of gastrointestinal disease (%)	2 (6.9%)	1 (10.0%)
History of *H. pylori* infection in gastric tissue (%)	2 (6.9%)	2 (20.0%)
Eradication of *H. pylori* in gastric tissue (%)	2 (6.9%)	2 (20.0%)

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
