# Peer review of "Distribution of Helicobacter pylori and Periodontopathic Bacterial Species in the Oral Cavity"

_biomedicines, 2020, doi:10.3390/biomedicines8060161_

Round 1

Reviewer 1 Report

The paper entitled “Distribution of Helicobacter pylori and periodontopathic bacterial species in the oral cavity” is an interesting work investigating the relationship between the colonization of oral cavity by H. pylori and periodontal disease. In particular, PCR was performed in oral cavity specimens of 39 patients with the ai to detect H. pylori and periodontal bacterial species. The results showed higher levels of red-complex species and lower levels of orange-complex species in patients with H. pylori. Moreover, P. gingivalis and T. denticola were associated with H. pylori.

The techniques utilized were appropriate and described with plenty details. This is a well-designed study with rigorous methods. The discussion is well-balanced and the statements are supported by the data. The study is on a timely subject, representing a proof of concept that periodontopathic pathogens may represent risk factors for colonization of the oral cavity by H. pylori.

I suggest some minor revision to improve the paper:

  • In Results section please report the number of cases and not only the percentages (if n < 100, the use of percentage implies a spurious impression of accuracy).
  • Regarding the role of H. pylori in gastric cancer development and the relationship between this pathogen and other bacteria of oral cavity, I suggest considering a recently published review about the relationship between oral microbiota and gastrointestinal tract cancer [1]. As the importance of the topic, I suggest adding some considerations regarding an even more important role of periodontal pathogens in human health.
  • Minor language corrections should be necessary.

[1] Mascitti, M., et al., Beyond Head and Neck Cancer: The Relationship Between Oral Microbiota and Tumour Development in Distant Organs. Front Cell Infect Microbiol, 2019. 9: p. 232.

Author Response

Response to Reviewer 1

The paper entitled “Distribution of Helicobacter pylori and periodontopathic bacterial species in the oral cavity” is an interesting work investigating the relationship between the colonization of oral cavity by H. pylori and periodontal disease. In particular, PCR was performed in oral cavity specimens of 39 patients with the ai to detect H. pylori and periodontal bacterial species. The results showed higher levels of red-complex species and lower levels of orange-complex species in patients with H. pylori. Moreover, P. gingivalis and T. denticola were associated with H. pylori.

The techniques utilized were appropriate and described with plenty details. This is a well-designed study with rigorous methods. The discussion is well-balanced and the statements are supported by the data. The study is on a timely subject, representing a proof of concept that periodontopathic pathogens may represent risk factors for colonization of the oral cavity by H. pylori.

I suggest some minor revision to improve the paper:

In Results section please report the number of cases and not only the percentages (if n < 100, the use of percentage implies a spurious impression of accuracy).

(Response)

As the reviewer suggests, we have added the number of cases to the Results section.

Regarding the role of H. pylori in gastric cancer development and the relationship between this pathogen and other bacteria of oral cavity, I suggest considering a recently published review about the relationship between oral microbiota and gastrointestinal tract cancer [1].

[1] Mascitti, M., et al., Beyond Head and Neck Cancer: The Relationship Between Oral Microbiota and Tumour Development in Distant Organs. Front Cell Infect Microbiol, 2019. 9: p. 232.

(Response)

Thank you for highlighting an important review article. The oral microbiota in cancer patients and healthy subjects is different, and specific oral microbiota may be involved in the development of cancers other than in the head and neck region. Alterations in the oral microbiota tend to be remarkable in cancer patients, especially in gastrointestinal tumor patients. In addition, some epidemiological studies have pointed to a relationship between periodontal disease and the risk of gastric cancer. We have added this information to the Introduction and Discussion sections of the revised manuscript.

As the importance of the topic, I suggest adding some considerations regarding an even more important role of periodontal pathogens in human health.

(Response)

As the reviewer points out, periodontal pathogens are associated with various systemic diseases, such as diabetes, cardiovascular disease, aspiration pneumonia, and preterm birth and low birth weight. We have added this information to the Introduction and Discussion sections of the revised manuscript.

Minor language corrections should be necessary.

(Response)

The revised manuscript has been edited and proofread by a native-English-speaking science editor from Edanz Group Ltd., as stated in the Acknowledgments section.

Reviewer 2 Report

In this article the authors analyzed the relationship between H. Pylori in the oral cavity and major periodontopathic bacterial species. There were 39 subject who underwent extraction of the third molar. 16 out of the 39 were impacted molars. Saliva, dental plaque and dental pulp were sampled from the subjects, on those with impacted teeth dental plaque was sampled from around the teeth. The patience characteristics were: Histories of systemic and gastrointestinal diseases, prior H. Pylori infection of gastric tissue, and prior eradication of H. Pylori from gastric tissue.

To detect H. Pylori from all the other bacterial species they used a nested PCR with a template bacterial DNA with H. Pylori-specific primers sets.

10 out 39 samples were defined as H. Pylori -positive subject. H. pylori was detected in 17.9% of dental plaque specimens, while 5.1% of saliva and 2.6% of dental pulp specimens were positive for H. pylori. Furthermore, 10 major periodontopathic bacterial species were detected in oral cavity specimens using a PCR method with species-specific sets of primers. In subjects with dental caries, those with H. pylori were more numerous than those without H. pylori, although the difference was not significant. Furthermore, periodontal pockets of 23 of 39 erupted teeth were measured, because the remaining 16 extracted teeth were impacted. Thus, the average periodontal pocket depths of H. pylori-positive and -negative specimens were 3.8 mm and 3.1 mm, respectively. They evaluated the relationship between the presence of H. pylori and the distribution of periodontopathic bacterial species in the oral cavity. At least four periodontopathic bacterial species were detected in most dental plaque specimens, with or without H. pylori, although a few were found in certain H. pylori-negative subjects.

These results suggest that highly pathogenic periodontopathic bacterial species may be related to colonization of the oral cavity by H. pylori.

One problem is about redactional problem.The manuscript is not good organized and it isn't in accord with journal's tamplate (the"materials and methods" in the end of manuscris, after the results and discussion) 

Another problem in this study is that the number of subjects was small and some of the subjects had impacted teeth that did not suffer any modifications from the oral cavity. To further the reaches of a study a larger pool of subjects, a variation of teeth, teeth that were in the oral cavity for at least 6 months and the samples of dental plaque and from dental pockets from different parts of the tooth should be used.

Another problem is that they did not have a control group. A control group that could furthermore indicate the prevalence of H. Pylori in patience with periodontal disease and it’s relation with specific periodontopathic bacterial species.

Author Response

Response to Reviewer 2

In this article the authors analyzed the relationship between H. Pylori in the oral cavity and major periodontopathic bacterial species. There were 39 subject who underwent extraction of the third molar. 16 out of the 39 were impacted molars. Saliva, dental plaque and dental pulp were sampled from the subjects, on those with impacted teeth dental plaque was sampled from around the teeth. The patience characteristics were: Histories of systemic and gastrointestinal diseases, prior H. Pylori infection of gastric tissue, and prior eradication of H. Pylori from gastric tissue.

To detect H. Pylori from all the other bacterial species they used a nested PCR with a template bacterial DNA with H. Pylori-specific primers sets.

10 out 39 samples were defined as H. Pylori -positive subject. H. pylori was detected in 17.9% of dental plaque specimens, while 5.1% of saliva and 2.6% of dental pulp specimens were positive for H. pylori. Furthermore, 10 major periodontopathic bacterial species were detected in oral cavity specimens using a PCR method with species-specific sets of primers. In subjects with dental caries, those with H. pylori were more numerous than those without H. pylori, although the difference was not significant. Furthermore, periodontal pockets of 23 of 39 erupted teeth were measured, because the remaining 16 extracted teeth were impacted. Thus, the average periodontal pocket depths of H. pylori-positive and -negative specimens were 3.8 mm and 3.1 mm, respectively. They evaluated the relationship between the presence of H. pylori and the distribution of periodontopathic bacterial species in the oral cavity. At least four periodontopathic bacterial species were detected in most dental plaque specimens, with or without H. pylori, although a few were found in certain H. pylori-negative subjects.

These results suggest that highly pathogenic periodontopathic bacterial species may be related to colonization of the oral cavity by H. pylori.

One problem is about redactional problem. The manuscript is not good organized and it isn't in accord with journal's tamplate (the "materials and methods" in the end of manuscris, after the results and discussion)

(Response)

We apologize that our article was not in the correct format for Biomedicines. In the revised manuscript, we have moved the Materials and Methods section to precede the Results section. In addition, we carefully checked that all parts of the manuscript complied with the journal guidelines.

Another problem in this study is that the number of subjects was small and some of the subjects had impacted teeth that did not suffer any modifications from the oral cavity. To further the reaches of a study a larger pool of subjects, a variation of teeth, teeth that were in the oral cavity for at least 6 months and the samples of dental plaque and from dental pockets from different parts of the tooth should be used.

(Response)

We totally agree with the reviewer’s comment, and large-scale studies using more subjects and samples taken from various teeth should be performed. Recent analyses of oral bacterial species, including periodontopathic bacteria, have led to important findings related to systemic diseases, such as gastric cancer, in addition to the known associations with diabetes and arteriosclerosis (Mascitti et al., Frontiers in Cellular and Infection Microbiology, 2019). Many of these studies involved small sample sizes. Systematic reviews and meta-analyses based on these papers have indicated promising associations (Mascitti et al., Frontiers in Cellular and Infection Microbiology, 2019). Therefore, we consider that our research may lead to clinically important novel findings. We have added this information to the Discussion section of the revised manuscript.

Periodontitis is gingival inflammation caused by bacterial infection of the periodontal tissues of tilted or impacted wisdom teeth, or unerupted teeth (Dhonge et al., International Journal of Dental and Medical Research, 2015). Therefore, we detected H. pylori from wisdom teeth that were diagnosed as impacted teeth, and H. pylori was detected in 3 of 16 (18.8%) impacted teeth. This result indicates that impacted wisdom teeth may be a possible reservoir for H. pylori colonization. We have added this information to the Discussion section of the revised manuscript.

Another problem is that they did not have a control group. A control group that could furthermore indicate the prevalence of H. Pylori in patience with periodontal disease and it’s relation with specific periodontopathic bacterial species.

(Response)

In the revised manuscript, subjects were divided into control (periodontal pocket of 3 mm or less) and periodontitis groups (periodontal pocket of 4 mm or more), according to the depth of the periodontal pocket of the extracted tooth. The distribution of H. pylori and periodontopathic bacterial species in these groups was analyzed and the results have been added to Supplementary Fig. 2.

There are many reports that compare the detection of H. pylori in healthy subjects (control) with that in periodontal disease patients. However, few studies have analyzed the oral bacteria of H. pylori-positive and H. pylori-negative oral cavities. Therefore, we compared the difference in the distribution of periodontal bacterial species in the presence or absence of H. pylori. We have added this information to the Discussion section of the revised manuscript.

Reviewer 3 Report

The work is poor, especially that presence of Helicobacter pylori was described already in many, many research.

Unfortunately, the main problem of this study is very low number of patients (39), and when we divide them into subgroups, also it is too low for statistical analysis. In only 23 periodontal pockets were only 5 positive samples with H. pylori. These numbers are to low for good statistics.

The name "Gram" e.g. Gram-positive, should be capitalized because it comes from the name of Hans Christian Gram.

In Table 2 it is unknown which primers are forward or reverse.

The conclusion may be misleading. Authors concluded that periodontopathic pathogens can be risk factors for colonization of the oral cavity by H. pylori. However, looking at the age of patients, H. pylori-positive samples were more common in older people. Periodontal changes, gingivitis and periodontitis are also more common in these people. At the same time, it is known that the occurrence of H. pylori increases with age, both in the stomach and in oral cavity. And this seems to be the main reason why the authors "discovered" a correlation between the occurrence of periopathogens and H. pylori. Moreover, periopathogens are not dominant in the stomach, but H. pylori infections are present. Therefore, it is doubtful that periopathogens were risk factor for colonization of the oral cavity by H. pylori. Simply with age, both types of bacteria are more common.

Author Response

Response to Reviewer 3

The work is poor, especially that presence of Helicobacter pylori was described already in many, many research.

(Response)

As the reviewer points out, there are a large number of studies showing that H. pylori is present in the oral cavity. However, it is difficult to specifically detect H. pylori from approximately 700 different bacterial species that can reside in the oral cavity, and the detection frequency of H. pylori varies from 0% to 100% (Robert et al., Periodontology 2000, 2013). In the present study, we present reliable data obtained using a highly accurate PCR method, which we recently established by designing primers in the common region of the genome sequence from 50 H. pylori strains (Ogaya et al., Journal of Medical Microbiology, 2015; Nomura et al., BMC Oral Health, 2018). In addition, most of the studies have detected only H. pylori in the oral cavity, and few have attempted to detect oral bacterial species at the same time. Therefore, we believe that our research builds on previous studies. We have added this information to the Introduction and Discussion sections of the revised manuscript.

Unfortunately, the main problem of this study is very low number of patients (39), and when we divide them into subgroups, also it is too low for statistical analysis. In only 23 periodontal pockets were only 5 positive samples with H. pylori. These numbers are to low for good statistics.

(Response)

We totally agree with the reviewer's comments, and the small sample size limits the statistical power of the study. To the best of our knowledge, only one study has focused on the relationship between H. pylori and oral bacteria including periodontopathic bacterial species, which analyzed 28 dental plaque samples from 14 subjects (Hu et al., Oncotarget, 2017). By referring to this paper, we decided to analyze three different samples: dental plaque, saliva and dental pulp from 39 subjects.

Recently, clinically important findings, such as those related to systemic diseases, have been made by the molecular analysis of oral bacterial species (Mascitti et al., Frontiers in Cellular and Infection Microbiology, 2019). Many of these studies involved small sample sizes. In addition, systematic reviews and meta-analyses using these papers have indicated possible associations (Mascitti et al., Frontiers in Cellular and Infection Microbiology, 2019). Therefore, we believe that our study may lead to novel important findings. We have added this information to the Discussion section of the revised manuscript.

The name "Gram" e.g. Gram-positive, should be capitalized because it comes from the name of Hans Christian Gram.

(Response)

The word “Gram” has been capitalized, as suggested.

In Table 2 it is unknown which primers are forward or reverse.

(Response)

We have indicated whether primers are forward or reverse in Table 2.

The conclusion may be misleading. Authors concluded that periodontopathic pathogens can be risk factors for colonization of the oral cavity by H. pylori. However, looking at the age of patients, H. pylori-positive samples were more common in older people. Periodontal changes, gingivitis and periodontitis are also more common in these people. At the same time, it is known that the occurrence of H. pylori increases with age, both in the stomach and in oral cavity. And this seems to be the main reason why the authors "discovered" a correlation between the occurrence of periopathogens and H. pylori. Moreover, periopathogens are not dominant in the stomach, but H. pylori infections are present. Therefore, it is doubtful that periopathogens were risk factor for colonization of the oral cavity by H. pylori. Simply with age, both types of bacteria are more common.

(Response)

Based on comments from the reviewers, we have revised the conclusions to "Taken together, these results suggest that H. pylori can co-exist with specific periodontopathic bacterial species, although interactions among the bacteria are not clear." As the reviewer points out, the incidences of both H. pylori and many periodontopathic bacterial species increase with age. Therefore, to clearly show the relationship between bacterial species, analyses using a larger number of specimens are needed. However, many studies have shown that periodontal disease is a risk factor for colonization of the oral cavity by H. pylori and for gastric cancer (Lauritano et al., Journal of Biological Regulators and Homeostatic Agents, 2015; Ahn et al., Cancer Causes and Control, 2012; Mascitti et al., 2019). Nevertheless, the periodontopathic bacterial species that co-exist with H. pylori were not known. Therefore, we aimed to identify differences in the periodontal bacterial species present in H. pylori-positive and H. pylori-negative oral specimens. We have added this information to the Discussion section of the revised manuscript.

Round 2

Reviewer 2 Report

In present form, the manuscript is ok.